# LRRC25 Inhibits IFN-γ Secretion by Microglia to Negatively Regulate Anti-Tuberculosis Immunity in Mice

**DOI:** 10.3390/microorganisms11102500

**Published:** 2023-10-05

**Authors:** Gang Sheng, Hongqian Chu, Huijuan Duan, Wenjing Wang, Na Tian, Dingyi Liu, Hong Sun, Zhaogang Sun

**Affiliations:** 1Beijing Chest Hospital Affiliated to Capital Medical University, Beijing 100000, China; shengg0622@163.com (G.S.); chuhongqian@bjxkyy.cn (H.C.); wenjing_wang96@foxmail.com (W.W.); tianna1998@163.com (N.T.); ldy20221@163.com (D.L.); 2Beijing Thoracic Tumor and Tuberculosis Institute, Beijing 100000, China; dhjhuijuan@163.com

**Keywords:** microglia, *Mycobacterium tuberculosis* (*Mtb*), LRRC25, ISG15, interferon gamma (IFN-γ)

## Abstract

Background: Leucine-rich repeat-containing protein-25 (LRRC25) can degrade the ISG15 gene in virus-infected cells and prevent overactivation of the type Ⅰ IFN pathway. However, the role of LRRC25 in bacterial infection is still unclear. In this pursuit, the present study aimed to explore the regulatory role and mechanism of LRRC25 in microglia infected with *Mycobacterium tuberculosis* in a mouse model. Methods: Q-PCR, WB, and cell immunofluorescence were employed to observe the change in LRRC25 in BV2 cells infected by H37Rv. Additionally, siRNA was designed to target the LRRC25 to inhibit its expression in BV2 cells. Flow cytometry and laser confocal imaging were used to observe the infection of BV2 cells after LRRC25 silencing. Q-PCR and ELISA were used to determine the changes in IFN-γ and ISG15 in the culture supernatant of each group. Results: Following H37Rv infection, it was observed that the expression of LRRC25 was upregulated. Upon silencing LRRC25, the proportion of BV2 cells infected by H37Rv decreased significantly. ELISA analysis showed that IFN-γ and ISG15 levels in cell culture supernatant decreased after H37Rv infection, while they significantly increased after LRRC25 silencing. Conclusions: This study provides evidence that LRRC25 is the key negative regulator of microglial anti-*Mtb* immunity. It exerts its function by degrading free ISG15 and inhibiting the secretion of IFN-γ, thereby improving the anti-Mtb immunity of BV2 cells.

## 1. Introduction

Leucine-rich repeat (LRR) proteins, including leucine-rich repeat-containing protein 25 (LRRC25), are known to be involved in cellular processes, such as apoptosis, autophagy, ubiquitination, nuclear mRNA transport, and neuronal development. These proteins play a significant role in modulating the host’s innate immunity, especially in innate immune receptors like nucleotide-binding oligo-domain (NOD)-like receptors (NLRs) and Toll-like receptors (TLRs) [1,2,3]. LRRC25 consists of 305 amino acids, with 4 leucine-rich repeats at the protein end, and has potential N-linked glycosylation sites. Additionally, LRRC25 contains an F-box domain that interacts with the E3 ubiquitin ligase, participating in ubiquitination biological processes [4]. Increasing evidence suggests that LRRC25 plays a crucial negative role in downregulating the inflammatory response caused by virus infection. Feng et al. demonstrated that LRRC25 interacts with p65/RelA, an upstream transcription factor of NF-KB, thereby promoting the autophagic degradation of P65/Rela to inhibit the transcription of NF-KB [5]. Du et al. showed that LRRC25 facilitates the autophagic degradation of RIG-l by binding ISG15, P62, and RIG-l to inhibit IFN-α/β expression mediated by RIG-l activation [6]. Through this mechanism, LRRC25 can prevent the overactivation of the IFN-α/β pathway and maintain the immune homeostasis in the host. Furthermore, the construction of an LRRC25 knock-down cell line significantly enhanced the cell’s ability to combat virus infection. Nonetheless, the function of LRRC25 in modulating the innate immunity during *Mycobacterium tuberculosis* (*Mtb*) infection remains unclear.

Tuberculous meningitis represents the most serious form of tuberculosis with high mortality rates [7]. When *Mtb* breaches the blood–brain barrier (BBB), it triggers an inflammatory reaction of the peripheral immune cells and resident immune cells within the brain region. Microglia, the primary macrophages in the central nervous system, are the main cellular targets in the *Mtb* infection in the central nervous system [8,9]. Microglia exert an innate cellular immunity by defending against pathogens and damaged neurons, as well as recognizing foreign invaders. They also play a crucial role in antigen presentation in the central nervous system and phagocytize pathogens [10,11,12]. Generally, the central nervous immune system is composed of microglia, astrocytes, and neurons that are known to limit the spread of *Mtb* in the brain. Upon *Mtb* invasion, the activated microglia release numerous cytokines and chemokines, contributing to the defense against central nervous system infection and neurogenesis-related pathogenesis [13,14]. Activated microglia differentiate into M1 (pro-inflammatory phenotype) and secrete pro-inflammatory cytokines, including IFN-γ, TNF-α/β, and IL-1α/β [15,16], as confirmed in the pneumococcal meningitis animal models [17,18]. However, excessive activation can elicit a nonspecific immune response and damage the nerve tissue [19]. Studies have identified LRRC25 as a potential risk gene for Alzheimer’s disease (AD) by sequencing the transcriptome of microglia and analyzing chromatin accessibility. It was found to play a key role in the occurrence and development of AD [20]. Given the important role of microglia in neuroinflammation, the present study aimed to investigate the LRRC25 in *Mtb*-infected microglia of mice (BV2 cells) and provide insights into the pathogenesis and treatment of tuberculous meningitis.

We explore the involvement of LRRC25 in mouse microglia infected by *Mtb*. The results showed a significant upregulation of LRRC25 expression in *Mtb*-infected microglia. By interfering with the expression of LRRC25 through siRNA transfection, the anti-infection ability of microglia against *Mtb* was significantly enhanced, highlighting the importance of LRRC25 as a molecule that negatively regulates the microglial susceptibility to *Mtb*. Moreover, evidence suggests that the negative regulatory role of LRRC25 may involve the degradation of ISG15 and the inhibition of IFN-γ secretion by cells. Overall, this study provides preliminary insights into the negative regulatory role of LRRC25 in neuromicroglial anti-*Mtb* immunity and confirms its function as a key negative regulatory molecule of ISG15, thereby reducing the anti-*Mtb* infection ability of microglia by suppressing IFN-γ release.

## 2. Materials and Methods

### 2.1. Cell Culture

Mouse microglia (BV2 cells), mouse astrocytes (C8-D1A cells), and mouse hippocampal neurons (HT22 cells) were procured from Wuhan Procell Life Science and Technology Company, China. Three cell lines were cultured in DMEM (Gibco, Grand Island, NY, USA) supplemented with 10% fetal bovine serum (PAN-Biotech, Aidenbach, Germany) and 1% penicillin/streptomycin solution (P/S) at 37 °C and 5% CO_2_. For further experiments, the resuscitated cells were passaged at least three times and underwent mycoplasma contamination detection.

### 2.2. Revival and Culture of H37Rv and GFP-H37Rv

The *Mtb* standard strains H37Rv (ATCC27294) and GFP-H37Rv (recombinant *Mtb*-H37Rv strain stably expressing GFP constructed with vector PSC301 (31851, Addgene, Watertown, MA, USA)) used in this study were obtained from the Beijing Tuberculosis Clinical Sample Database of Beijing Chest Hospital affiliated with Capital Medical University, China. The frozen bacterial liquid was thawed and mixed thoroughly. Subsequently, 50 μL of bacterial liquid was transferred to 25 mL culture medium. GFP-H37Rv was cultured in 7H9 liquid medium containing hygromycin at 37 °C in the dark for 21 days. H37Rv was cultured in 7H9 liquid medium at 37 °C in the dark for 21 days.

### 2.3. Transfection and Silencing

siRNA, designed and synthesized by GenePharma (Suzhou, China), was dissolved in DEPC water and transfected with Lipofectamine RNAi Max (Invitrogen, Shanghai, China) in a 3:1 ratio, following the protocol provided by Invitrogen (https://www.thermofisher.cn/order/catalog/product/13778075, accessed on 1 October 2023, MAN00007825 Rev.1.0, 2013). Before siRNA transfection, BV2 cells were cultured for 24 h in DMEM without antibiotics, and no change in the culture medium was required during transfection. The respective siRNA sequences are shown in Appendix A.

### 2.4. Mtb Infection in BV2 Cells

To prepare the bacterial suspension, 25 mL of the bacterial liquid was centrifuged at 10,000 revolutions per minute (RPM) for 5 min, and the supernatant was discarded. The bacterial pellet was resuspended in 5 mL of antibiotic-free culture solution. BV2 cells were infected with H37Rv at various multiplicities of infection (MOIs)—1, 5, or 10. The infected BV2 cells were incubated at 37 °C for 4 h and then washed three times with PBS.

### 2.5. Quantitative Real-Time PCR (Q-PCR)

After the cell split, RNA was extracted by using an RNA rapid extraction kit (Vazyme, Nanjing, China). Reverse transcription was performed by using a Fast Quant RT Kit with gDNase, KR106 (Tiangen, Beijing, China), and a real-time fluorescence quantitative PCR detection was performed by using PowerUp SYBR Green Master Mix (Applied Biosystems, Waltham, MA, USA). QuantStudio™ 12k Flex Real-Time PCR System (Applied Biosystems, Waltham, MA, USA) was employed for amplification. The PCR reaction volume was 20 μL, and the PCR reaction procedure was as follows: pre-denaturation: 95 °C, 5 min; amplification: 95 °C, 30 s; 58 °C, 30 s; and 72 °C, 30 s; for 50 cycles. The mRNA relative expression of target gene was calculated by 2^−∆∆Ct^ method. A gene is considered unexpressed when its Ct value exceeds 35. Primers were synthesized by Sangon Biotech (Shanghai, China), and the sequences are shown in Appendix A.

### 2.6. Western Blot Analysis

RIPA lysis buffer (Beyotime Biotechnology, Shanghai, China) containing 1 mM phenylmethanesulfonyl fluoride (PMSF) (Servicebio, Hubei, China) was used to lyse the cells. Protein was quantified by BCA kit (Beyotime Biotechnology, Shanghai, China). The ratio of loading buffer to sample was 1: 4, and the sample was denatured by boiling at 95 °C for 5 min. The sample was separated by 10% SDS–PAGE (sample size: 15 μg/well, 120 V, 20 min, then 160 V, 40 min. The nitrocellulose filter membrane (NC) (Solarbio Life Sciences, Beijing, China) was used for membrane transfer at 400 mA for 40 min. Samples were incubated with primary antibodies overnight at 4 °C in the dark (LRRC25 antibody: Santa Cruz; 1:200). The next day, after washing the NC membrane with TBST (1×) 3 times, the membrane was incubated with the secondary antibody (HRP-labeled goat anti-mouse: Servicebio; 1:6000) for 1 h. Then, ECL chromogenic solution (Epizyme Biomedical Technology, Shanghai, China) was used for visualization. The Western blot setup was provided by Bio-Rad (Hercules, CA, USA). Electrophoresis buffer, transfer buffer, and TBST buffer were purchased from Servicebio (Hubei, China).

### 2.7. Laser Confocal Microscopy Inspection

GFP-H37Rv was used to infect the cells (2 × 10^5^ cells/well) in the confocal dish for 4 h at an MOI of 5. After washing the cells with PBS 3 times, 4% paraformaldehyde was added and fixed for 15 min. After washing with PBS 3 times, Hoechst staining (1:1000 dilution with sterile PBS) was performed for 15 min, and then the cells were resuspended after washing with PBS 3 times. Images were observed and acquired under the laser scanning confocal fluorescence microscope (FV1000, Olympus, Tokyo, Japan) at an image magnification of 600×.

### 2.8. Cellular Immunofluorescence

GFP-H37Rv was used to infect the cells (2 × 10^5^ cells/well) in the confocal dish for 4 h at an MOI of 5. Cells were washed with PBS 3 times, and 4% paraformaldehyde was added and then fixed at room temperature for 10 min. Then, the cells were washed with PBS 3 times and made permeable with 0.5% Triton X-100. Subsequently, the cells were incubated overnight at 4 °C with LRRC25 antibody (LRRC25 antibody: Santa Cruz; 1:200). After washing the cells with PBS 3 times, the cells were incubated with a fluorescent antibody (Abcam: ab150115; 1:200) for 1 h. After three washes with PBS, the cells were stained with Hoechst for 5 min. Finally, images were obtained using a laser scanning confocal fluorescence microscope (FV1000, Olympus, Tokyo, Japan) at a magnification of 600 times.

### 2.9. Flow Cytometry

GFP-H37Rv was used to infect the cells (2 × 10^5^ cells/well) in the 6-well cell culture plates for 4 h at MOI of 5. The culture solution containing the bacterial suspension was discarded and washed with PBS 3 times. Trypsin was used to elute cells for 20s, and DMEM without antibiotics was added into each well. Subsequently, the cell suspension was transferred to a centrifuge tube and centrifuged at 1000 RPM for 5 min, and the supernatant was discarded. After washing the cells with PBS 3 times, the suspension was centrifuged at 1000 RPM for 5 min. A 4% paraformaldehyde was used to fix cells at 4 °C for 12 min and was centrifuged at 1000 RPM for 5 min. Cells were washed with PBS and centrifuged at 1000 RPM for 5 min, and supernatant was discarded. The cells were flowed through the cell analysis platform (Guava, TX, USA) at a medium speed, and a total of 10,000 cells were measured. The cells were first gated by forward scattered light (cell size) and side scattered light (granularity) to select against cellular debris. Further analysis was conducted on the delineated cell population. One vertical gate was set, one selecting infected cells (Green-B positive) and one selecting uninfected cells (Green-B negative), only the cells infected by GFP-H37Rv showed Green-B positive.

### 2.10. Pathological Section

The paraffin sections were deparaffinized using xylene and absolute ethanol to achieve transparency. Hematoxylin dye solution was applied to the sections for 3–5 min and subsequently washed with PBS. By employing differentiation solution, the slices were dehydrated in 85% and 95% gradient alcohol for 5 min, respectively, and stained with a solution of eosin for 5 min. The sections were then sequentially placed in absolute ethanol Ⅰ, anhydrous ethanol Ⅱ, and anhydrous ethanol Ⅲ for 5 min each. Subsequently, the sections were soaked in xylene Ⅰ and xylene Ⅱ for 5 min. After sealing with neutral gum, the slices were observed using a microscope (Olympus, Tokyo, Japan).

### 2.11. ELISA

H37Rv was removed from the cell culture supernatant by using the 0.45 μm filter (Merck Millipore, Darmstadt, Germany) and centrifuged at 2500 RPM for 20 min. Cell culture supernatants were assayed by mouse IFN-α, mouse IFN-γ, mouse IL-1β, and mouse ISG15 ELISA kits (Mei Mian, Yancheng, China) according to the manufacturer’s instructions, and the absorbance of wells was measured at 450 nm using Multiskan SkyHigh (Thermo Fisher, Waltham, MA, USA).

### 2.12. Establishment of Mouse Model with Tuberculous Meningitis

All mice employed for the study were SPF-grade female C57BL/6 mice purchased from Vital River (Beijing, China) with an average weight of 20 g (8 to 10 weeks old). Animals were tested after adapting to the environment for one week. H37Rv in the logarithmic growth phase was used to prepare a 1 × 10^7^ CFU/mL bacterial suspension, and mice were randomly divided into two groups, an infected and an uninfected group (12 mice/group). Each mouse in the infected group was injected with 200 μL (1 × 10^7^ CFU/mL) of the above bacterial suspension intravenously through the tail vein, and 200 μL of normal saline was injected intravenously into each mouse in the uninfected group through the tail vein. Two weeks after infection, the mice were treated with the same injection again. Neurobehavioral measurements and follow-up experiments were performed at 0 weeks, 2 weeks, 4 weeks, and 6 weeks after infection (3 mice/group). The method of sacrifice was neck-breaking. After sacrifice, the brain tissue was divided into cerebrum and cerebellum, which were soaked in 4% paraformaldehyde tissue fixative solution (Servicebio, Hubei, China). Unfixed brain tissue was homogenized in 3 mL of physiological saline. The homogenized brain tissue was then diluted 10 times with physiological saline, and 100 μL was evenly spread on a 7H10 culture plate to count the CFU of brain tissue.

### 2.13. Cell Viability Detection

BV2 cells were inoculated with 1 × 10^4^ cells per well into a 96-well plate in an antibiotic-free DMEM medium and cultured at 37 °C and 5% CO_2_ for 24 h. Transfection was performed according to the procedure stated in Section 2.3, and then infection was performed according to the procedure stated in Section 2.4. Subsequently, the culture medium was removed, and 100 mL of DMEM containing 10% cell counting kit—8 (CCK—8, NCM Biotech, Suzhou, China) was added. Absorbance was measured at 450 nm at 30 min, 1 h, and 4 h using an ELISA plate reader (Thermo Fisher, Waltham, MA, USA), respectively, and repeated 5 times. Three independent experiments were performed for the assay. Cell viability was expressed as follows:Cell viability = (Absorbance of Test wells/Mean absorbance of Control wells) × 100%

### 2.14. Data Analysis

IBM SPSS 9.0 (obtained at https://www.ibm.com/cn-zh/spss, accessed on 1 October 2023) was employed for statistical analysis. Firstly, we tested whether the data followed a normal distribution and variance homogeneity. Then, we selected the appropriate statistical method. Flow cytometry data analysis was carried out using FlowJoTM10 (https://www.flowjo.com/solutions/flowjo/downloads, accessed on 1 October 2023).

### 2.15. Ethical Approval

This study was reviewed by the Ethics Committee of Beijing Chest Hospital, affiliated to Capital Medical University. Approval Code: YJS-2021-103. Approval Date: 26 November 2021.

## 3. Results

### 3.1. LRRC25 Is Involved in the Anti-Mtb Natural Immunity of Microglia

To investigate the best MOIs, BV2 cells were infected with H37Rv with a GFP fluorescent label (GFP-H37Rv) at MOIs of 1, 5, and 10. The results indicated that when the MOI was 1, there were fewer infected cells compared to those at MOIs of 5 and 10. However, at an MOI of 10, some cells experienced cell death. Considering these findings, an MOI of 5 was selected (Figure 1a). To confirm the impact of *Mtb* infection on LRRC25 in microglia, BV2 cells were infected with the standard strain of *Mycobacterium tuberculosis* (H37Rv). After 4 h of infection, the mRNA and protein levels of LRRC25 were assessed using Q-PCR and Western blotting. The results revealed a significant increase in the transcription and protein expression of LRRC25 in the infected BV2 cells (Figure 1b,c). Moreover, cellular immunofluorescence images demonstrated a significant upregulation of LRRC25 expression in the infected group (Figure 1d). To determine whether LRRC25 was specifically expressed in the main central nervous system cells, C8-D1A cells and HT22 cells were cultured and infected with H37Rv. The expression levels of LRRC25 mRNA were then examined after 4 h of infection. The results showed that BV2 cells exhibited high expression levels of LRRC25, which were significantly upregulated after *Mtb* infection. Conversely, LRRC25 was not expressed in C8-D1A cells, and its expression in HT22 cells was low and downregulated after infection (Appendix A). These results strongly suggest that LRRC25 plays a vital role in the innate immunity of BV2 cells against *Mtb* infection. To investigate the expression of LRRC25 in the anti-*Mtb* immunity of microglia in vivo, a mouse model of tuberculous meningitis was established. Brain tissue sections were stained and analyzed for LRRC25 expression at 2, 4, and 6 weeks. After 4 weeks of infection, the mice exhibited notable neurological symptoms, along with clear inflammatory cell infiltration and sporadic lesions observed in brain tissue sections (Figure 1e). Double-labeled immunofluorescence analysis of IBA-1 and LRRC25 expression in mice with tuberculous meningitis showed a significant increase in LRRC25 expression compared to the control group. The IBA-1 marker was localized to the BV2 cell membrane, while LRRC25 was found around the nucleus (Figure 1f). Immunohistochemical images further confirmed the increased expression of LRRC25 in the mouse model (Appendix A). Overall, these findings indicate that LRRC25 exhibits increased expression at both the transcriptional and translational levels following H37Rv infection of BV2 cells. Additionally, LRRC25 was highly expressed in BV2 cells after *Mtb* infection, indicating its important role in the innate immune response against *Mtb* infection. Furthermore, in the mouse model of tuberculous meningitis, LRRC25 expression was significantly upregulated, suggesting its involvement in the anti-*Mtb* immune response of microglia.

### 3.2. LRRC25 Deficiency Enhances the Anti-Mtb Immunity of Microglia

To explore the physiological function of LRRC25 in the anti-*Mtb* immunity of microglia, we designed and synthesized three LRRC25-specific small interfering RNA (siRNA) sequences (L1, L2, and L3) to silence the LRRC25 expression. We delivered them into BV2 cells using Lipofectamine RNAiMAX as a vector to knock down LRRC25. The silencing efficiency of the siRNA was assessed at 48 h and 72 h. Q-PCR results revealed that L1 exhibited the highest silencing efficiency at both time points. Furthermore, the silencing efficiency at 72 h was significantly greater than that at 48 h (Figure 2a). Western blot analysis showed no significant difference in the protein levels at 48 h. However, at 72 h, the LRRC25 expression was significantly reduced in the L1 and L2 groups, with L1 displaying the most effective silencing effect (Figure 2b). Therefore, we selected L1 as the interference sequence and transfected it for 72 h to effectively silence LRRC25. To further determine the effect of LRRC25 deletion on the anti-*Mtb* immunity of BV2 cells, we infected BV2 cells with H37Rv for 4 h at an MOI of 5 after 72 h of L1 transfection. After cells were lysed, mRNA and protein were extracted for Q-PCR and WB. We found that after 72 h of transfection, Q-PCR showed a significant difference between LRRC25 expression in the test group and the control group (Figure 2c). Furthermore, WB analysis showed a notable decrease in LRRC25 protein levels (Figure 2d). These observations indicated the successful silencing of LRRC25. Subsequently, flow cytometry and laser confocal knock-down assays were conducted. The flow cytometry analysis demonstrated a 79% increase in the proportion of negative cells (uninfected with *Mtb*) in the LRRC25 group compared to the control group (NC) (GAPDH group: 7.89%, NC group: 7.17%, and LRRC25 group: 12.9%) (Figure 2e). Additionally, laser confocal imaging demonstrated a significant reduction in the number of *Mtb*-infected BV2 cells of the LRRC25 group (Figure 2f). The susceptibility of BV2 cells to *Mtb* was significantly reduced. These results showed that LRRC25 deficiency could significantly enhance the anti-tuberculosis infection ability of BV2 cells. To ensure that the decrease in the proportion of infected cells following LRRC25 knock-down was not due to cell death, we assessed cell viability before and after transfection and infection. The data revealed no significant change in cell viability among the different groups after adding CCK-8 for 30 min, 1 h, and 4 h (Appendix A).

### 3.3. LRRC25 Is an Important Negative Regulatory Signal for Microglia to Release IFN-γ

The above results show that microglia are different from other central nervous cells; microglia highly express LRRC25 and become susceptible to tuberculosis after *Mtb* infection. Next, we attempted to determine whether LRRC25 could inhibit the release of ISG15 in the extracellular space by mediating the degradation of ISG15 and then inhibit the release of IFN-γ to negatively regulate the anti-tuberculosis immunity of BV2 cells. We first assessed whether the BV2 cells were effectively activated by quantifying the mRNA levels of IL-1β, a representative indicator of BV2 cell activation, and measured the levels of IL-1β in cell culture supernatants of the group before and after transfection and before and after infection. We found that interference of LRRC25 alone could increase IL-1β (48 h vs. 72 h, *p* < 0.001), but *Mtb* infection can significantly increase the IL-1β levels (48 h vs. 72 h, *p* < 0.001). Similarly, we also observed a significant increase in the expression of IL-1β in the brain of the mouse model for tuberculosis meningitis (Appendix A). This indicates that BV2 cells can be effectively activated by *Mtb,* leading to a series of biological effects (Figure 3a). After confirming the activation of BV2 cells, we measured the extracellular release of IFN-α, IFN-γ, and ISG15 from BV2 cells before and after *Mtb* infection. We found that IFN-α in cell supernatant increased significantly, while IFN-γ and ISG15 decreased significantly after BV2 cells were infected with H37Rv (Figure 3b). To further confirm whether the decrease in ISG15 and IFN-γ is caused by the activation of LRRC25, we quantified the mRNA of ISG15 in the four groups of cells before and after transfection and before and after infection. We then measured the ISG15 level in each group by ELISA (Figure 3c). The results showed that compared with the untransfected group, the secretion level of ISG15 increased significantly after 72 h of transfection, and the degradation of ISG15 was successfully rescued by silencing LRRC25. Even after the H37Rv infection, the effect still persisted. To determine whether silencing LRRC25 promoted the release of IFN-γ, we next measured the IFN-γ levels in each group (Figure 3d). We found that after LRRC25 was silenced, a higher level of IFN-γ was detected in the cell culture supernatant of the L1 group compared with that of the NC group. These results indicate that LRRC25 can inhibit the release of IFN-γ by preventing ISG15 from being released to negatively regulate the anti-tuberculosis immunity of microglia. Furthermore, we found that knock-down of LRRC25 could increase the level of IFN-α, which is consistent with previous reports [6].

## 4. Discussion

The role of LRRC25 has been extensively studied in viral infection. LRRC25 is known to combine with ISG15, P62, and RIG-l in a noncovalent way to form a complex, mediate the selective autophagy of ISG15, and inhibit the type I IFN signaling pathway. Studies have shown that LRRC25 can enhance the antiviral immunity of cells [6]. However, the role of LRRC25 in bacterial infection, particularly *Mtb* infection, has not been reported yet. In this study, we further explored the role of LRRC25 in *Mtb* infection. We found that when microglia were infected by *Mtb*, the expression of LRRC25 was significantly increased at the gene and protein levels, as verified by immunofluorescence experiments. Moreover, we observed a significant increase in LRRC25 expression in the brain region of mice with tuberculous meningitis at the animal level. To investigate the relationship between LRRC25 and the anti-tuberculosis immunity of microglia, we used siRNA to interfere with the expression of LRRC25. Surprisingly, we found that the low expression of LRRC25 actually enhanced the anti-*Mtb* immunity of microglia, suggesting its role as a key negative regulator. However, further research revealed that the infection of *Mtb* could effectively activate the expression of IFN-α but inhibit the secretion of IFN-γ. Knocking down LRRC25 can save the secretion of IFN-γ and ISG15. Thus, our study demonstrates that LRRC25 negatively regulated the anti-*Mtb* immunity of microglia, and the mechanism was possible by promoting the degradation of ISG15 and reducing the production and secretion of IFN-γ. However, we still need more direct and in-depth evidence to further explore the relationship between LRRC25 and IFN-γ.

Numerous studies have focused on the subtle relationship between the *Mtb* infection and the type I IFN pathway. Prabhakar et al. described in detail the regulatory effect of *Mtb* infection on IFN-α/β. The data showed that macrophages had a low level of self-secretion of IFN-α/β under normal conditions, but there was a negative feedback mechanism to counteract it [21]. Although highly pathogenic *Mycobacterium* infection significantly increased the transcription and expression of type Ⅰ IFN in macrophages, its negative feedback pathway was also continuously activated to inhibit the expression of the type Ⅰ IFN signal pathway. Although this inhibition is not enough to limit the production of type Ⅰ IFN from cells, it played a crucial role in *Mtb* evading the host immunity. Our study also found that the early stage of *Mtb* infection can effectively activate the type I IFN pathway. Furthermore, studies have shown that the ESX-1 protein secretion system, such as CFP10 and ESAT-6, is necessary for macrophages to produce and release type Ⅰ IFN, and the activation of TANK-binding kinase 1 (TBK1) is the key link of this pathway. We also found that after silencing LRRC25, stimulating microglia with ESAT-6 and CFP10 could increase the transcription of LRRC25 again, and the expression of ISG15 still persisted (Appendix A), suggesting that ESAT-6 and CFP10 may be the main pathogenic elements involved in the action pathway of LRRC25 [22].

ISG15 is an intracellular ubiquitin-like protein (UBL) that is induced by type I IFN. The expression level of ISG15 increases noticeably when the type I IFN pathway is activated. Conversely, human cells that lack ISG15 exhibit a continuous increase in the expression of type Ⅰ IFN stimulating product (ISG). In addition, they also exhibit a significant reduction in virus susceptibility. This suggests that ISG15 may be a potent inhibitor of type I IFN. One possible mechanism is that ISG15 can stabilize the protein structure of USP18, thereby inhibiting the continuous activation of the type I IFN signaling pathway [23,24]. When ISG15 is absent, the type I IFN signaling pathway remains activated and offers virus resistance. Our study supports this inference. In the BV2 cells infected with H37Rv for 4 h, IFN-α remained at a high level, while ISG15, a transcription product of type I interferon, decreased significantly. This indicates that intracellular LRRC25 may degrade ISG15 via autophagy. Furthermore, ISG15 can exist in cells in two forms: bound and free. The bound form can be covalently coupled with a variety of intracellular proteins to form ISGylated proteins [25]. The free form can be secreted by a variety of cells, such as neutrophils, monocytes-macrophages, and lymphocytes. The free ISG15 can approach the CD3+ T lymphocytes and natural killer (NK) cells and combine with lymphocyte function-related antigen-1 (LFA-1) on its surface to induce T lymphocytes and natural killer cells to produce IFN-γ [26,27,28]. Additionally, stimulation increases the expression of CD11a protein on the cell surface of mouse microglia, facilitating their mediation in migration and adhesion [29,30,31]. Genetic studies have shown that defects in the ISG15 gene can result in Mendelian susceptibility to *Mycobacterium* disease (MSMD) due to decreased excretion of ISG15, leading to IFN-γ deficiency, and increased susceptibility to *Mtb*, BCG, and environmental *Mycobacterium* (EM) [32,33,34]. Our research reveals a significant reduction in the extracellular secretion of ISG15 in the *Mtb*-infected microglia, which could inhibit the production of IFN-γ and render BV2 cells susceptible to *Mtb*.

IFN-γ, type II interferon, is produced and secreted by T cells, NK cells, and dendritic cells. Secreted IFN-γ can act on macrophages to polarize them to the M1 type and stimulate the release of cytokines, such as IL-1β and TNF-α [35,36,37,38]. Jung et al. showed that IL-1β increased significantly after macrophages were infected with *Mtb,* effectively inhibiting the invasion of *Mtb* into BV2 cells [39]. Under certain conditions, macrophages can produce and secrete IFN-γ in an autocrine manner under certain conditions, such as direct contact with pathogens, and the secreted IFN-γ can act on macrophages again to activate macrophages, affecting their state and function [21]. However, further verification is required to explain the source of IFN-γ in the cell culture supernatant in our study. While IFN-γ mainly plays an antiviral role, the specific mechanism of the direct regulation of the anti-tuberculosis immunophenotype of cells remains unclear and needs further exploration.

IL-12 is an inducer of IFN-γ production, and IFN-γ can also promote the production of IL-12. At present, the phenomenon of expression of IL-12 in the blood or cells of TB patients is contradictory [40,41,42]. However, more studies support the finding that patients with active *Mtb* have higher levels of IL-12 [43,44]. We measured IL-12 in BV2 cells before and after H37Rv infection and found that the IL-12 in the cell supernatant increased significantly after infection. Our findings showed a significant increase in the secretion of IL-12 levels in the cell supernatant after infection. Conversely, when LRRC25 was knocked down, there was a significant increase in the secretion of IL-12 in the cell supernatant. Additionally, the expression of IL-12 was significantly lower after transfection for 72 h compared to 48 h (Appendix A). The specific reasons for this result need to be further evaluated. Combined with the determination results of IFN-γ, we speculate that IL-12, as a component of innate anti-tuberculosis immunity, gets activated following an *Mtb* infection. However, it is insufficient to counteract the inhibitory effect of LRRC25 when a large number of *Mtb* infect BV2 cells, resulting in a downward trend in IFN-γ levels in the infected group.

It has been suggested that LRRC25 can promote the degradation of p65/RelA and inhibit the activation of the NF-kB pathway. In order to determine whether LRRC25 plays a role in the NF-κB pathway, we detected the classic molecule p65/RelA activated by the NF-κB pathway. Initially, we measured the expression level of p65/RelA mRNA in BV2 cells before and after tuberculosis infection and observed that tuberculosis infection did not affect the expression of p65 (Appendix A). However, we found that the knock-down of LRRC25 significantly reduced the expression level of p65/RelA (Appendix A); nevertheless, the interaction between LRRC25 and p65/RelA remains unclear and requires further exploration.

Based on the experimental data mentioned earlier, we propose a possible mechanism to elucidate how LRRC25 negatively regulates the anti-*Mtb* immunity of microglia: when *Mtb* or its cell components, such as CFP10 and ESAT-6, penetrate the BBB and reach microglia, they induce the synthesis and release of ISG15. Subsequently, ISG15 is released into the extracellular space and acts on microglia, activating them and releasing IFN-γ and IL-12. However, concomitantly, the transcription and expression of LRRC25 get enhanced. LRRC25 can degrade ISG15 by noncovalent coupling, preventing its overactivation and release and maintaining the stability of ISG15 in cells (Figure 4).

## 5. Conclusions

The present study showed that LRRC25 negatively regulates the anti-tuberculosis immunity of microglia, and one possible mechanism is the activation of the type Ⅰ IFN pathway when *Mtb* or its bacterial components, such as EC, penetrate the blood–brain barrier and reach the microglia. Simultaneously, LRRC25 also gets activated and binds to and degrades ISG15, resulting in a decrease in ISG15 released out of cells, which inhibits the secretion of IFN-γ by cells. Consequently, the ability of microglia to eliminate tuberculosis is weakened.

## Figures and Tables

**Figure 1 microorganisms-11-02500-f001:**
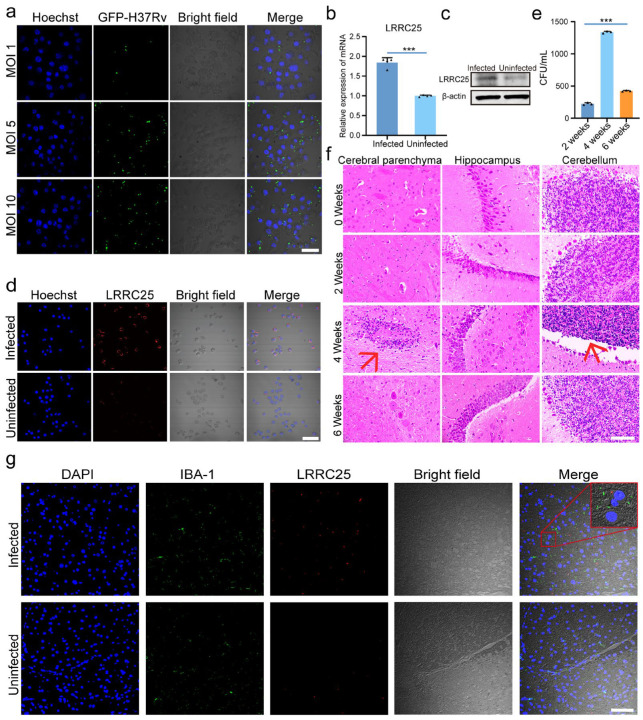
Increase in the transcription and translation of LRRC25 in H37Rv-infected BV2 cells. (**a**) The laser scanning confocal images GFP-H37Rv infects BV2 cells with MOIs of 1, 5, and 10 (n = 3). (**b**) Relative expression of LRRC25 mRNA after H37RV infection of BV2 cells for 4 h (n = 4), Student’s *t* test. (**c**) Western blot images of LRRC25 after H37RV infection of BV2 cells for 4 h (n = 3). (**d**) Cellular immunofluorescence image of LRRC25 H37RV-infected BV2 cells for 4 h. Scale bar, 100 μm. (**e**) H37Rv CFU count of mice brain tissue at 2 weeks, 4 weeks, and 6 weeks (n = 3), Kruskal–Wallis test. (**f**) HE staining images of the brain regions of mice infected with tuberculosis at 0, 2, 4, and 6 weeks (red arrows show the inflammatory cell infiltration areas. There were three mice in each group. Scale bar, 50 μm). (**g**) Immunofluorescence image of cerebellum of mice infected with H37Rv at 6 weeks; the nucleus is blue, IBA-1 is green, LRRC25 protein is red, and the scale is 50 μm (n = 3). The figure shows a representative result. *** *p* < 0.001.

**Figure 2 microorganisms-11-02500-f002:**
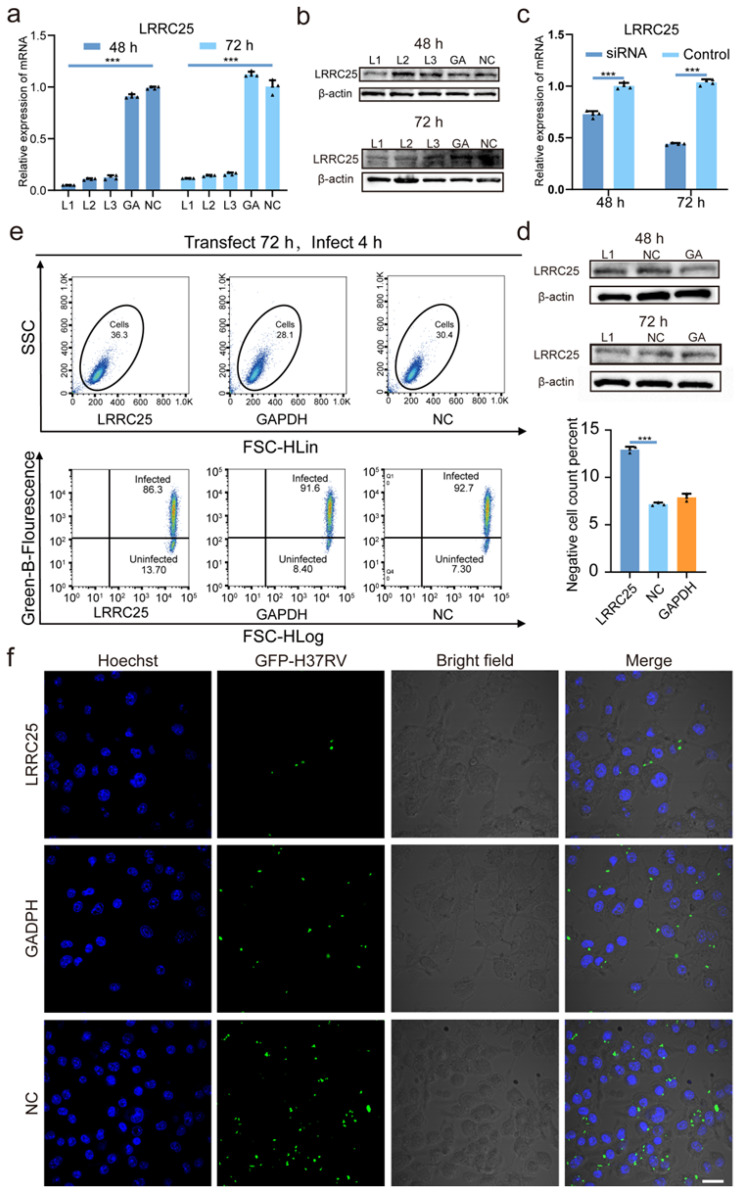
Interference of the translation of LRRC25 using siRNA of LRRC25: L1, L2, and L3. (**a**) Relative expression of LRRC25 mRNA after transfection of BV2 cells for 48 h and 72 h (n = 4), One-way ANOVA. (**b**) Western blot images of LRRC25 after transfection of BV2 cells for 48 h and 72 h (n = 3). (**c**) Relative expression of LRRC25 mRNA in BV2 cells after being transfected for 48 h and 72 h and infected by GFP-H37Rv for 4 h at an MOI of 5 (n = 4), Student’s *t* test. (**d**) Western blot images of LRRC25 in BV2 cells after being transfected for 48 h and 72 h and infected by H37Rv for 4 h at an MOI of 5 (n = 3). (**e**) Histogram and percentage statistics of uninfected BV2 cells in flow cytometry (n = 3), One-way ANOVA, Ex post facto comparison -LSD. (**f**) Confocal images of infected BV2 cells. Among them, the nucleus was stained and marked with Hoechst (blue), and GFP-H37Rv was marked with green fluorescent protein (green). Scale bar, 50 μm. BV2 cells were transfected with L1 for 72 h and infected with GFP-H37Rv for 4 h at an MOI of 5. GA is the abbreviation for GAPDH, and NC is the abbreviation for Negative Control. *** *p* < 0.001.

**Figure 3 microorganisms-11-02500-f003:**
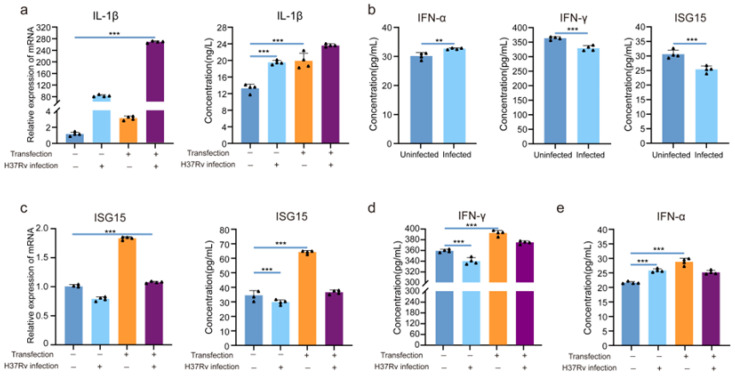
LRRC25 prevents BV2 cells from releasing ISG15 and IFN-γ. (**a**) Relative expression of IL-1β mRNA of BV2 and secretion level of IL-1β in the cell supernatant of each group. (**b**) Levels of IFN-α, IFN-γ, and ISG15 in the cell culture supernatant before and after H37Rv infection of BV2 cells. (**c**) Relative expression of ISG15 mRNA and the expression of ISG15 in the cell supernatant in each group. (**d**) Expression of IFN-γ in the cell supernatant in each group. (**e**) Expression of IFN-α in the cell supernatant in each group. Bar graphs show the mean ± SD from four independent experiments (n = 4). Student’s *t* test for figure (**b**) and Two-way ANOVA test for figure (**a**,**c**,**d**). BV2 cells were transfected with L1 for 72 h and infected with GFP-H37Rv for 4 h at an MOI of 5. ** *p* < 0.01, and *** *p* < 0.001.

**Figure 4 microorganisms-11-02500-f004:**
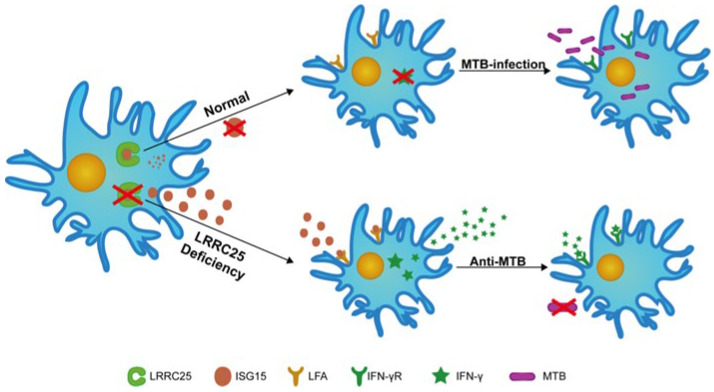
Illustration of the negative regulation of anti-tuberculosis immunity in microglia by the LRRC25-ISG15-IFN-γ axis.

## Data Availability

The data presented in this study are available in the Appendix A.

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
