# Peer review of "LRRC25 Inhibits IFN-γ Secretion by Microglia to Negatively Regulate Anti-Tuberculosis Immunity in Mice"

_microorganisms, 2023, doi:10.3390/microorganisms11102500_

Round 1

Reviewer 1 Report (Previous Reviewer 1)

I appreciate that the authors addressed my comments; however, several mistakes in the manuscript make the review of the manuscript difficult because it cause a distraction, and the focus is mainly on mistakes and not on the significance of content. Authors should improve and polish their writing, correcting the mistakes. There are some points in the manuscript that the authors need to address to improve the quality.

In Methods, authors used different units for centrifugation, they should use only one (Relative Centrifugal Force (RCF) or Revolutions Per Minute (RPM).

 siRNA sequences, and Primers sequences, it could be better in tables in supplementary information.

In the 2.9 section (Flow cytometry), the authors described the procedure for flow cytometry, but it is incomplete, there is no methodology for flow cytometry; they omitted which molecules were evaluated, stain (antibodies used), data acquisition, gating strategy, etc. 

They described only infection and recovery of infected cells, flow cytometry procedure should be included.

There are some mistakes of grammar in the main text. For example:

Line 14- 15 " were designed for targeting the LRRC25 was designed to...

Line 50 "  it tiggers...

Line 300 "Thereore..

Line 372 "used Wilcoxon-ruskal-Wallis test, a, c and d used Tow-way ANOVA test...

Etc..

Also, there are several mistakes in Figure 3 (subheadings of axes).

Overall, the font size on the axes is very small in Figure 2, and inside the do plots.

Author Response

Reviewer 2 Report (New Reviewer)

Tuberculosis (TB) is a global health problem throughout the world. One of the deadliest complications is TB meningitis. In the paper under review, the authors investigate the role of LRRC25 in anti-TB immunity. The work is relevant and the methods used are appropriate to the task.

There are a few minor comments:

1. What is meant by ddCt (Figure 1b)? dCt - housekeeping gene standardization. What is ddCt? The graph shows mRNA levels from infected and uninfected cells separately and it remains unclear what exactly is being shown.

2. Similar question to Figures 2 and 3

3. Fig. 3b The graphs of IFNg and ISG15 do not need a break in the scale.

4. Describe the kit CCK-8. Information not available in Materials and methods.

Round 2

Reviewer 1 Report (Previous Reviewer 1)

The quality of figures must be improved, and in some of them, the text is still small (letter size) and it can not read clearly (for example, Fig2 e).

There are mistakes in y-axis titles (Fig 3), and also, in the legend of Fig 3 (line 388). Is it a Kruskal-Wallis test? and the name of the Two-way  ANOVA test is incorrect. The authors should correct these mistakes.

In Table S3, authors show the CT mean and the comparison between conditions. Does gene expression was assessed by the delta delta Ct method as they mention in the methods section? If is it, they need to compare relative expressions (where RQ= 2-∆∆Ct), not mean of CT.

In 

English need to be polished, because there are still some mistakes.

Author Response

This manuscript is a resubmission of an earlier submission. The following is a list of the peer review reports and author responses from that submission.

Round 1

Reviewer 1 Report

The authors demonstrated LRRC25 mediates microglial anti-tuberculosis immunity in a mice model, LRRC25 a negative regulator that promotes, ISG15 degradation and inhibiting the secretion of IFN-γ. It is promising for be considered as a potential therapeutic target for tuberculous meninges. However, there are several comments needed to be addressed before publication.

1)     In introduction, the significance of microglia and its relevance in immunity against the tuberculous infection should be better illustrated, also statistics of bacterial meningitis could be included; and some references could be more recent, for example ref 1, 6, 7, 11; because some of them are a Review.

2)     In the figure 2 caption, why authors chose Mann-Whitney U test for Relative expression of LRRC25 (panel a), and Student’s t test for Relative expression of LRRC25 (panel a), if both have a sample size n=3; moreover, this sample size is too small to test normality distribution of data.

3)     For confocal microscopy, authors should indicate the sample size (experimental replicates, not only the visual fields). At least 3 experimental replicates should be considered.

4)     The material and methods section should be rewrite, because there are several grammatical mistakes in the following procedures:

2.3. Establishment of BV2 infection cell model, 2.5. Western blot, 2.8. Flow cytometry, 2.9. Pathological section, and 2.10. ELISA (sentences should be rephrased; they are too long and unclear).

Also, some parts of discussion section should be rewrite.

“Our research has further verified these conclusions. We found that the changes in IFN-γ and IL-1….”

“affecting their state and function of their function….”

5)     Authors reported that they did not find significant differences in IFN-α, IFN-γ and ISG15 in the cell culture supernatant after BV2 infection (Figure. 3b); however, there is a significant decrease IFN-γ and ISG15, correct this sentence.

6)     There are several mistakes overall of document that should be corrected, e.g., “INF-γ”, “denatured protein sample 15 μ g/well, 15 120 V,20 min”

7)     Abbreviatures should be homogenize, like ISG15.

8)     The part  after Figure 4, should be removed "This section may be divided…."

Some descriptions of the materials and methods section are not described correctly, the reading difficult to flue.

Reviewer 2 Report

Sheng et al. explore the role of Leucine-rich repeat containing 25 (LRRC25) in Mycobacterium tuberculosis (Mtb) infected mice and a cell line of microglia (BV2) cells. Given that there is some evidence that LRRC25 negatively regulates viral-induced immune responses and in microglia specifically, the authors hypothesized that silencing LRRC25 could improve the anti-bacterial functions of microglia. They subsequently utilized siRNA-induced silencing of LRRC25 in Mtb H37RV-infected BV2 cells to conclude that LRRC25 deficiency promotes anti-Mtb immunity through upregulation of ISG15 and IFNg signaling. While the report is potentially interesting to the Mtb and neuroinflammation community, there are several major issues with the manuscript that need to be addressed before it can be considered for publication. 

For all figures involving immunostainings for microscopy of the BV2 cells, a microglial marker should be used. In figure 1 for example, IBA-1 is only used in panel F. While a more microglia-specific marker other than IBA-1 would be preferred, IBA-1 should suffice given that BV2 is a well-validated cell line. Nonetheless, in 1F, the staining looks poor (too spotty and difficult to see the outline of the cell). Moreover, it looks as if many of the cells would be negative for IBA-1 in the culture and it is impossible to assess which cells are potentially expressing LRRC25. The staining in 1D for LRRC25 looks better than in 1F but there is no microglial marker in the panel. 

For the In vivo pictures (Figure 1E), it would be beneficial to both show the GFP-H37RV and an IBA-1 marker along with the H&E staining.

The flow cytometry in figure 2 should include the full gating strategy that include a Live-Dead cell panel and markers (CD45intCD11b+ or TMEM119+) to demonstrate that they are in fact microglia.

Why did the authors use 2 hours of infection time followed by 24 hours of incubation time when establishing the BV2 infection model, and 4 hours of infection time without additional incubation for the rest of the experiments? In the case of cytokine measurement from the cell culture supernatant, the authors do not specify incubation length following infection.

Western blots in figure 2 are not quantified and do not seem to reflect the trends observed at the mRNA level. Why is this the case?

The data with ISG15 and IFNg are mostly correlational. It would increase the impact of the paper to have mechanistic data to demonstrate if/why/how LRRC25 prevents the release of IFNg from microglia.

There is inconsistency in the data. For example, in Figure 3B shows a marked reduction in IFNg production by BV2 cells upon infection. In contrast, Figure 3D shows a slight increase in IFNg concentration between in the non-transfected groups (dark blue versus orange bar) following H37Rv infection. Throughout the paper, the authors were making the case that infection reduces IFNg in the supernatant. This is potentially troubling and may suggest a greater N is needed before making conclusions. 

Why do uninfected BV2 cells secrete IFNg (Figure 3B) without stimulation? 

It seems LLRC25 silencing increased the secretion of IL-1b, ISG15, iFNg and IFNa production in the absence of infection. How do the authors explain this?

How did the BV2 cell viability change following transfection and/or infection?

The authors hypothesize that LRRC25 silencing might enhance antimicrobial activity of microglia via enhancing IFNg production by these cells. However, to prove this, they show flow cytometry and IHC data with reduced intracellular bacterial burden after an incubation time as short as 4 hours (Fig 2E,F). Would this short time be enough for Mtb uptake, cell activation, IFNg production, bacterial killing and the complete loss of bacterial fluorescence from the cells? Given the short time and the huge effect seen in the IHC, other possible factors should also be considered (reduced bacterial uptake, change in BV2 cell viability etc.).

For bar graphs, each data point should be shown along with SD error bar. N of 5-6 with multiple independent experiments for each assay would be much more ideal than N of 3 and add validity to the data.

For statistical analyses, Mann-Whitney test cannot be used when comparing more than 2 groups. ANOVAs must be used to assess whether there is significance when 3 or more groups are involved and a Two-way ANOVA when there are two factors (infection and transfection) involved while evaluating a response variable (Figure 3 A, C, D, E for example).

There are several typos throughout the manuscript, I also recommend extensive English editing. 

For example, see the following sentence in the introduction: "Compared with the nonknockout strain, the anti-virus infection ability of the cell is significantly enhanced by constructing LRRC25 knockout cell line for the virus infection experiment."

Instead of the phrasing "Feng et al. thought...", Du et al. thought...", when describing previous research, I recommend using the verbs shown, proved, described, etc.

"GADPH" vs correct: GAPDH

The Materials and Methods section contains passive and active voice, present and past tense, e.g. "H37Rv was used to infect processed cells (2×10^5 cells/ well) in the 6 well cell culture plates for 4 h at an MOI of 5. Discarding the culture solution containing bacterial, then add 2ml PBS to each well and wash it with sterile PBS for 3 times. Using trypsin to eluted cells for 20s and add DMEM without antibiotics into each well."

Reviewer 3 Report

The manuscript microorganisms-2529790 titled “LRRC25 inhibits IFN-γ secretion by microglia to negatively 2 regulate anti-tuberculosis immunity in mice” by Gang Sheng et al., is a study focused on the role of LCRR25, a protein previously described as a player in the development of Alzheimer Disease (AD), in bacterial infection and innate inmunnity, specifically in a murine model of Mycobacterium tuberculosis (Mtb) infection.

The manuscript is, in my opinion, not acceptable in its present form. The conclusions presented are not supported by the showed results. The rationale of the study is not well explained, the models of infection are not well described and the formatting of the manuscript is extremely poor. I am even surprised about the fact that this version passed the first editorial filters and were selected as elegible for peer-review.

 Major concerns

The authors (lines 55-61) try to present the rationale under the investigation of LRRC25, which has been previously linked to AD, in a totally different context, in this case Mtb infection. However, in AD, inflammation is generally considered secondary to the damage due to β-amyloid deposition. Granulocytes have been proposed as important players in this disease. All these pieces of data do not explain the rationale that drove the authors to explore the role of LRRC25 in bacterial infection and, in particular, in a murine microglia Mtb infection model.

Another importan flaw of the study is the poor characterization of both the cell culture and murine infection models. Specifically in the murine model, since Meningeal TB is usually secondary to a previous pulmonary or intestinal TB infection, the authors should present evidence supporting the presence of Mtb in the neural tissue to prove that the model is useful. Finally, but also a major concern, the manuscript is poorly formatted and needs an exhaustive grammar/style revision since, in its present form is extremely difficult to follow and understand.

Some examples:

-Lines 37-41: "The authors thought”. The authors should use a more scientific term instead, such as “demonstrated”, “proposed”, etc..

-Lines 44-45:  The sentence is confusing,  rewrite

-Line 63: Specify “microglia cell line”.

-Lines 85, 112,141, 198, 209, 281: Missing or extra spaces between words.

-Lines 168, 179: Avoid presenting Material and Methods as in a protocol book and use the proper verbal tense.

-Line 225: Fig. S1 is missing in the version I downloaded. Link on the manuscript is not working.

-Line 250: Fig S2 is missing in the version I downloaded. Link on the manuscript is not working.

-Line 272. Use “delivered” instead of transported.

-The results and discussion are presented alternating cell culture experiments and mice tissue data and the rationale to jump from one model to another is quite poor. This is a very difficult section to follow and the whole part should be rewritten.

-Lines 361-363: Another example of the poor formating of the present manuscript.

-The original western blot images are missing in the folder they were suppose to be.

The manuscript is poorly formatted and needs an exhaustive grammar/style revision since, in its present form is extremely difficult to follow and understand.